# Epidemiology and Outcome of Primary Central Nervous System Tumors Treated at King Hussein Cancer Center

**DOI:** 10.3390/cancers17040590

**Published:** 2025-02-09

**Authors:** Maysa Al-Hussaini, Abdallah Al-Ani, Justin Z. Amarin, Sarah Al Sharie, Mouness Obeidat, Awni Musharbash, Amer A. Al Shurbaji, Ahmad Kh. Ibrahimi, Abdellatif Al-Mousa, Nasim Sarhan, Nisreen Amayiri, Rula Amarin, Tala Alawabdeh, Qasem Alzoubi, Dima Abu Laban, Bayan Maraqa, Khaled Jamal, Asem Mansour

**Affiliations:** 1Department of Cell Therapy and Applied Genomics, King Hussein Cancer Center, Amman 11941, Jordan; 2Department of Pathology and Laboratory Medicine, King Hussein Cancer Center, Amman 11941, Jordan; bm.11034@khcc.jo; 3Office of Scientific Affairs and Research, King Hussein Cancer Center, Amman 11941, Jordan; abdallahalany@gmail.com (A.A.-A.); justinzamarin@gmail.com (J.Z.A.); sarahalsharie2000@gmail.com (S.A.S.); kjamal@khcc.jo (K.J.); 4Department of Surgery, King Hussein Cancer Center, Amman 11941, Jordan; mo.15790@khcc.jo (M.O.); awnimusharbash@gmail.com (A.M.); dr.shurbaji@hotmail.com (A.A.A.S.); 5Department of Radiation Oncology, King Hussein Cancer Center, Amman 11941, Jordan; aibrahimi@khcc.jo (A.K.I.); aalmousa@khcc.jo (A.A.-M.); nsarhan@khcc.jo (N.S.); 6Department of Pediatrics, King Hussein Cancer Center, Amman 11941, Jordan; namayiri@khcc.jo; 7Department of Internal Medicine, King Hussein Cancer Center, Amman 11941, Jordan; ramarin@khcc.jo (R.A.); ta.11388@khcc.jo (T.A.); 8Department of Radiology, King Hussein Cancer Center, Amman 11941, Jordan; qa.13630@khcc.jo (Q.A.); da.11945@khcc.jo (D.A.L.); amansour@khcc.jo (A.M.)

**Keywords:** central nervous system tumors, epidemiology, survival rates

## Abstract

This study investigates the epidemiology and outcome of primary central nervous system (CNS) tumors treated at the King Hussein Cancer Center (KHCC) in Jordan. We analyzed data from over 2000 patients diagnosed between 2003 and 2019. Our findings show that gliomas are the most common type of CNS tumor, and survival rates differ significantly between age groups and genders. Adult patients had lower survival rates compared to pediatric patients, and male patients had poorer outcomes compared to females. We also observed that tumors with specific genetic mutations had better survival rates. These findings align with global trends but highlight the growing burden of CNS tumors in the region. This research is crucial for understanding the unique aspects of CNS tumors in Jordan and emphasizes the need for region-specific cancer policies and improved diagnostic tools.

## 1. Introduction

Primary central nervous system (CNS) tumors are a heterogeneous group of benign and malignant neoplasms that arise from the tissues of the brain and spinal cord. As of 2021, the WHO CNS5 adjusted the taxonomy of primary CNS classifications, adding newly recognized entities, and removing obsolete tumor types [1]. According to the GLOBOCAN 2022 estimates, malignant CNS tumors accounted for 1.7% of new cancer diagnoses and 2.57% of deaths due to cancer globally [2]. However, the incidence is disproportionate in children and adolescents; where malignant CNS tumors are the second and third most common type of cancer, respectively [3]. In addition, the burden of malignant CNS tumors varies regionally, especially between developed and developing countries. For instance, estimates of burden are higher in Jordan compared with worldwide estimates. In 2018, malignant CNS tumors accounted for an estimated 3.2% of new cancer diagnoses and 4.6% of deaths due to cancer in Jordan, ranking 10th and 6th, respectively [4]. The national cancer registry in Jordan (the Jordan Cancer Registry) registered 58,788 new cases between 2000 and 2013, and malignant CNS tumors accounted for 2096 of the cases (3.6%)—ranking 9th overall [5].

The most common pathological entities of CNS tumors in children aged 0–14 years are gliomas (predominantly pilocytic astrocytoma) and embryonal tumors (predominantly medulloblastoma) [6]. In adolescents and young adults aged 15–39 years, the most common tumors are pituitary tumors, followed by meningiomas and astrocytomas [7]. In older adults (ages 40+ years), the most common type is meningioma, followed by gliomas (predominantly glioblastoma) and pituitary tumors [8]. According to a multicenter one-year study in Jordan, the most common histologies across all age groups are meningioma, glioblastoma, astrocytoma, and pituitary neuroendocrine tumors (PitNETs), formally known as pituitary adenomas [9]. Generally, among the different histologies of CNS tumors, the outcomes of ependymoma, pilocytic astrocytoma, and oligodendroglioma are relatively favorable; the 5- and 10-year relative survival point estimates are 80.6–87.8% and 63.8–84.5%, respectively. On the other hand, the outcomes of glioblastoma, and anaplastic astrocytoma are relatively poor; the 5- and 10-year relative survival point estimates are 5.4–29.9% and 2.7–20.8%, respectively [10].

The behavior of a specific CNS tumor entity varies widely with multifactorial attributes. For instance, the outcomes of malignant CNS tumors vary by age, biological sex, and ethnicity [10,11,12]. Interestingly, the incidence rates of CNS tumors vary from region to region, likely due to distinct environmental and genetic risk factor profiles. Therefore, regional studies may help clarify the complex etiologies of CNS tumors [11]. Given the above, our aim was to study the epidemiology and outcome of primary CNS tumors in patients managed at a comprehensive cancer center in Jordan.

## 2. Materials and Methods

### 2.1. Study Design and Settings

We performed a retrospective chart review of all newly diagnosed patients with a primary CNS tumor who were managed at the King Hussein Cancer Center (KHCC) between July 2003 and June 2019. King Hussein Cancer Center is the only comprehensive cancer center in Jordan, serving roughly 60% of all patients with cancer in Jordan [13]. We included all patients with a primary CNS tumor (ICD-O code 70–72 and 75), who were registered in the institutional Cancer Registry. We excluded patients who started treatment at another institution. We retrieved all data available in the registry. The registry collects and curates the following data for each patient: age, biological sex, anatomic site of tumor, histology of tumor, grade of the tumor, survival, and date of last follow-up. Our primary outcomes of interest were the 1-year, 2-year, and 5-year overall survival (OS). Overall survival (OS) was calculated from the date of diagnosis to the date of death or last follow-up. The Department of Civil Status and Passports was contacted to update the status of included patients.

Once the data were retrieved, one of the authors (M.A.H) performed a reclassification of the histology and grade of the tumors based on the WHO 2021 Classification of Tumors of the CNS System (CNS5), by reviewing the pathology and, when needed, the radiology reports. This included reclassifying the tumor groups. For example, the “gliomas, glioneuronal tumors, and neuronal tumors” group was created to include the six described tumor families including adult-type diffuse gliomas, pediatric-type diffuse low-grade gliomas, pediatric-type diffuse high-grade gliomas, circumscribed astrocytic gliomas, glioneuronal and neuronal tumors, and ependymal tumors. Also, all previous “obsolete” nomenclatures of diffuse astrocytoma, including protoplasmic astrocytoma, were grouped under astrocytoma. Other obsolete types including oligoastrocytoma were, when possible, reclassified into either astrocytoma or oligodendroglioma based on the result of ancillary studies in the pathology report including p53 immunostain and 1p/19q FISH results. New types that have not been identified as separate entities before were generated. For example, diffuse gliomas that occupy the midline structures including thalami, brainstem, and spinal cord were reclassified as diffuse midline gliomas. Grades were updated as per the new classification. For example, myxopapillary ependymomas were re-graded as grade-2, instead of grade-1, tumors. The descriptive term “anaplastic” was removed and the tumor was only assigned the corresponding grade, grade-3. As per the WHO classification, all Roman numbers were replaced with Arabic numbers to designate the grade. Moreover, IDH-1 (performed by immunohistochemistry using IDH-1 p.R132H clone), has been introduced since 2014 and the adult-type diffuse glioma were classified as IDH- mutant or wildtype.

### 2.2. Data Analysis

We used R (version 4.3.3) to perform data analysis. First, we stratified the data set by age group (<18 years or ≥18 years). Next, we computed summary statistics to describe each stratum as well as the full sample. Then, for each type of tumor, we plotted age group-specific survival curves using the Kaplan–Meier method and compared the curves using the log-rank test. We used the Benjamini–Hochberg method to correct for multiple testing and interpreted values of *p* ≤ 0.05 to indicate statistical significance. Throughout the report, we present numerical data according to Cole’s recommendations and in the following format: absolute counts alongside relative counts (the latter within parentheses) and means alongside standard deviations (the latter within parentheses) [14].

### 2.3. Ethical Considerations

The Institutional Review Board of the King Hussein Cancer Center approved the study protocol (20KHCC125).

## 3. Results

### 3.1. General Characteristics

We accessed data from the institutional cancer registry and identified 3351 records of primary CNS tumors. We excluded 1107 records of patients who were not nationals of Jordan, 48 duplicate records or records of second primary CNS tumors, and 102 records for other reasons. The final study population comprised 2094 patients, all of whom had complete records.

The median age at diagnosis was 33 years (IQR, 12–52 years). The number of pediatrics and adults was 652 (31.1%) and 1442 (68.9%), respectively. Males outnumbered females 1145 (54.7%) to 949 (45.3%). Figure 1 describes the sex distribution across different age groups.

The most common tumor site was the supratentorium (*n* = 1241 [59.3%]), followed by the infratentorium (*n* = 540 [25.8%]), meninges (*n* = 272 [13.0%]), and cranial nerves (*n* = 41 [2.0%]). Of all tumors, 719 (34.3%) were grade 1, 263 (12.6%) were grade 2, 224 (10.7%) were grade 3, and 888 (42.4%) were grade 4. Figure 2 and Figure 3 show the site distribution and grade per age group, respectively. 

The five most common groups of tumors were “gliomas, glioneuronal tumors, and neuronal tumors” (*n* = 1200 [57.30%]), followed by meningiomas (*n* = 261 [12.5%]), embryonal tumors (*n* = 234 [11.2%]), tumors of the cranial and paraspinal nerves (*n* = 58 [2.8%]), and tumors of the sellar region (*n* = 58 [2.8%]). The counts and proportions of all groups, types, and subtypes are listed in Appendix A. Figure 4 shows the distribution of CNS tumors, classified by WHO tumor group, per age group.

Table 1 shows tumors stratified by biological sex. Of 20 tumor types diagnosed at least 10 times, 13 types were more likely to occur in males, while 6 were more likely to occur in females. Germinoma, hemangioma, diffuse midline glioma, pediatric-type diffuse low-grade gliomas, medulloblastoma, adult-type diffuse gliomas, and glioneuronal and neuronal tumors were at least 1.2 times more likely to occur in males. On the other hand, meningiomas were more likely to occur in females. Table 2 also lists tumor entities, as per the WHO classficaition, stratified by location of origin.

### 3.2. Survival Analysis

The median survival for the entire cohort was 97 months (95CI; 81–112). The median survival was 71 months (95CI; 57–82) for males, while it was 139 months (95CI; 109–210) for their female counterparts (*p* < 0.001) [Figure 5]. Survival was worse for adult patients than their pediatric counterparts (76 months vs. 266 months; *p* < 0.001) [Figure 6]. When categorized by diagnosis period, there were no differences between those diagnosed prior to 2012 and those diagnosed in 2012 onwards (81 months vs. 91 months; *p* = 0.120) [Appendix A. Hazard of death was significantly worse for males (HR: 1.33; 95%CI: 1.18–1.52) and better for pediatrics (HR: 0.77; 95%CI: 0.67–0.88) after adjusting for diagnosis period.

For the included WHO groups, the OS rates are presented in Appendix A. Hematolymphoid tumors, tumors of the pineal region, melanocytic tumors, and “gliomas, glioneuronal tumors, and neuronal tumors” had the worst 1-, 2-, and 5-year OS rates.

#### 3.2.1. Adult Cohort

##### General Survival

The 1-, 2-, and 5-year OS rates for the adult cohort were 79% (77–81%), 66% (64–69%), and 54% (51–56%), respectively. There were significant differences in OS per biological sex across the aforementioned cohort (*p* < 0.001) (Appendix A).

##### Survival per WHO Groups and Types

“Gliomas, glioneuronal tumors, and neuronal tumors” comprised the majority of adult tumors (*n* = 819 [56.8%]), followed by meningiomas (*n* = 250 [17.3%]). The 1-, 2-, and 5-year OS rates for “gliomas, glioneuronal tumors, and neuronal tumors” were 67% (64–70%), 47% (43–50%), and 33% (39–36%), respectively.

Conversely, the 1-, 2-, and 5-year OS rates for meningiomas were 98% (96–100%), 95% (92–97%), and 84% (78–89%), respectively. Among the adult cohort with meningiomas, anaplastic meningioma had the worst OS (Appendix A). However, post hoc analysis shows no statistically significant OS differences between entities of meningioma. Table 3 shows the 1-, 2-, and 5-year OS for the included adult cohort. Survival metrics for WHO groups stratified per diagnosis period are presented in Appendix A.

##### Survival per WHO Types

Of 819 “gliomas, glioneuronal tumors, and neuronal tumors”, approximately 86.6% (*n* = 709) are adult-type diffuse gliomas. Tumor types included glioblastoma (61.9% [*n* = 439]), astrocytoma, grade-3 (13.3% [*n* = 94]), astrocytoma, grade-2 (12.4% [*n* = 88]), oligodendroglioma, grade-2 (5.9% [*n* = 42]), and oligodendroglioma, grade-3 (4.6% [*n* = 33]). Thirteen cases remained as oligoastrocytoma and could not be further classified.

The 1-, 2-, and 5-year OS rates for adult-type diffuse glioma types are present in Appendix A. Glioblastoma demonstrated significantly worse OS than astrocytoma, grade-3 (*p* < 0.001); oligodendroglioma, grade-3 (*p* < 0.001); astrocytoma, grade-2 (*p* < 0.001); and oligodendroglioma, grade-2 (*p* < 0.001). When analyzed per diagnosis period, glioblastoma maintained a significant survival difference to astrocytoma, grade-3; oligodendroglioma, grade-3; astrocytoma, grade-2; and oligodendroglioma, grade-2 (all *p* < 0.05) across the <2012 and ≥2012 periods (Appendix A).

##### Adult-Type Diffuse Glioma; IDH-Mutant vs. Wildtype

A separate sub-group analysis on cases where IDH mutations were examined demonstrated the following. Overall, tumors with IDH mutations had a survival advantage over wildtype cases (IDH-mutant 1-year OS, 89% [82–97%] vs. IDH-wildtype 1-year OS, 60% [52–70%]; *p* < 0.001) (Figure 7).

For patients with astrocytoma, there were significant survival differences between IDH-mutant and IDH-wildtype tumors (IDH-mutant 1-year OS, 91% [81–100%] vs. IDH-wildtype 1-year OS, 63% [46–85%]; *p* = 0.049), irrespective of the grade (Figure 8).

However, when analyzed by grade, IDH mutation does not result in survival differences across astrocytomas of either grade-2 or grade-3 variants (*p* = 0.085 and 0.300, respectively) (Figure 9).

For patients with glioblastoma, there were no significant survival differences between IDH-mutant and IDH-wildtype tumors (IDH-mutant 1-year OS, 78% [61–100%] vs. IDH-wildtype 1-year OS, 57% [48–69%]; *p* = 0.200) (Figure 10). When stratified by tumor type and by IDH mutation, there were significant survival differences between oligodendroglioma, IDH-mutant astrocytoma, and IDH-wildtype astrocytoma (Oligodendroglioma 1-year OS, 96% [89–100%] vs. IDH-mutant astrocytoma 1-year OS, 91% [81–100%] vs. IDH-wildtype astrocytoma 1-year OS, 63% [46–85%]; *p* < 0.001) (Figure 11).

#### 3.2.2. Pediatric Cohort

The 1-, 2-, and 5-year OS rates for the pediatric cohort were 79% (75–82%), 69% (65–72%), and 62% (59–66%), respectively. There were no differences in OS per biological sex across the aforementioned cohort (*p* = 0.800) (Appendix A). After adjusting for the diagnosis period, there were no differences in survival between the biological sexes.

##### Survival per WHO Groups and Families

While “gliomas, glioneuronal tumors, and neuronal tumors” were the most prevalent among pediatrics (*n* = 381 [58.4%]), the cohort had embryonal tumors as its second most prevalent tumor (*n* = 173 [26.5%]). Meningiomas only comprised 1.7% of all pediatric cases. The 1-, 2-, and 5-year OS rates for “gliomas, glioneuronal tumors, and neuronal tumors” were 75% (71–80%), 66% (62–71%), and 60% (55–65%), respectively. Figure 12 shows the survival differences between families of “gliomas, glioneuronal tumors, and neuronal tumors”. OS rates for pediatric-type diffuse high-grade gliomas were significantly lower than all other tumor families including circumscribed astrocytic gliomas, ependymal tumors, glioneuronal and neuronal tumors, and pediatric-type diffuse low-grade gliomas (all *p* < 0.001). Similarly, pediatric-type diffuse low-grade gliomas had worse OS than circumscribed astrocytic gliomas (*p* = 0.002), and glioneuronal and neuronal tumors (*p* = 0.044). Finally, ependymal tumors had worse OS than glioneuronal and neuronal tumors (*p* = 0.040).

On the other hand, the 1-, 2-, and 5-year OS rates for embryonal tumors were 77% (71–84%), 65% (58–72%), and 58% (51–67%), respectively. Among this group of tumors, medulloblastoma presented with favorable 1-, 2-, and 5-year OS rates (83% [78–90%], 74% [67–81%], and 67% [59–75%], respectively). Medulloblastoma had significantly the most favorable prognosis in comparison to atypical teratoid/rhabdoid tumors (*p* < 0.001), CNS embryonal tumors, NOS (*p* = 0.017), and embryonal tumors with multilayered rosettes (*p* < 0.001). Table 4 shows the 1-, 2-, and 5-year OS for the groups and types within the gliomas and embryonal tumors families among pediatrics. Appendix A demonstrates survival differences in “gliomas, glioneuronal tumors, and neuronal tumors” when stratified by diagnosis period.

##### Survival per WHO Tumor Types

Of 381 “gliomas, glioneuronal tumors, and neuronal tumors”, 41.21% were circumscribed astrocytic gliomas (*n* = 381), followed by pediatric-type diffuse high-grade gliomas (36% [*n* = 137]), ependymal tumors (14.4% [*n* = 55]), glioneuronal and neuronal tumors (4.7% [*n* = 18]), and pediatric-type diffuse low-grade gliomas (3.7% [*n* = 14]). Of 137 pediatric-type diffuse high-grade gliomas, 54.7% are diffuse midline gliomas (*n* = 75). At one year, DMG (32%; 23–45%) had the worst survival percentage in comparison to other entities, followed by glioblastoma off the midline (55%; 42–71%).

## 4. Discussion

To the best of our knowledge, this is one of the few presentations of a hospital-based registry of primary CNS tumors across the MENA region. Across 16 years, a total of 2094 Jordanian patients with primary CNS tumors were diagnosed and treated at the KHCC. The most common tumor groups were “gliomas, glioneuronal tumors, and neuronal tumors”, meningiomas, and embryonal tumors. “Adult-type diffuse gliomas” was the most common tumor family across the entire cohort. Survival analysis demonstrated significant differences at both the biological sex and age levels. Among types of “gliomas, glioneuronal tumors, and neuronal tumors”, adult-type diffuse gliomas and pediatric-type diffuse high-grade gliomas demonstrated the worse survival. Finally, sub-group analysis in the adult cohort on cases where IDH mutations were examined demonstrated that tumors with IDH mutations had a survival advantage over wildtype cases.

In comparison with international literature, only a number of large-scale databases provided individual-level statistics. These include the Central Brain Tumor Registry of the United States (CBTRUS) report, EUROCARE-5, and the VIGICANCER report [15,16,17,18,19]. The CBTRUS statistical report of 2015–2019 demonstrated that meningiomas (41.4%), gliomas, glioneuronal tumors, and neuronal tumors (25.0%), and tumors of the sellar region (17.9%) are the most common CNS tumors across both sexes, irrespective of age group and behavior (i.e., benign vs. malignant) [15]. As for the survival data, the report shows that meningiomas had a median survival of 185 months vs. 257 months in our report. We reported higher median survival for “diffuse” astrocytoma (CBTRUS: 61 vs. KHCC: 81), “anaplastic” astrocytoma (CBTRUS: 20 vs. KHCC: 28), and glioblastoma (CBTRUS: 8 vs. KHCC: 13). Conversely, median survival for oligodendroglioma was worse at KHCC (CBTRUS: 199 vs. KHCC: 104), where it was similar in “anaplastic” oligodendroglioma (CBTRUS: 103 vs. KHCC: 103). In another report, the registry shows that among malignant CNS tumors, gliomas, glioneuronal tumors, and neuronal tumors were the most common tumor families (82.1%) [16]. Embryonal tumors and meningiomas were only 2.7% and 2.1% of the malignant CNS tumors. Among non-malignant CNS tumors, meningiomas (55.2%) and tumors of the sellar region (25.1%) were the most common, with 92% 5-year all-ages survival rates for non-malignant tumors.

Due to a lack of access to the EUROCARE-6 data, the EUROCARE-5 reports, which cover the period between 2000 and 2007, were utilized for comparison [18,19]. It should be noted that the EUROCARE database only provides statistics for neuroepithelial CNS variants only. Among adults, glioblastoma was the most common specified neuroepithelial tumor (49.4%), followed by oligodendrogliomas/oligoastrocytomas (8.8%), ependymal tumors (1.5%), and embryonal tumors (1.1%) [18]. We reported relatively higher 5-year all-ages survival across four aforementioned entities (glioblastoma, 9.9% vs. 6.3%; oligodendrogliomas/oligoastrocytomas, 57.1% vs. 39.9%; ependymal tumors, 73% vs. 58.4%; and embryonal tumors, 70% vs. 35.3%, from KHCC vs. EUROCARE-5 report, respectively).

Among pediatrics, the limitations of EUROCARE-5 are more illuminated. Due to the system of classification used in the report (i.e., international classification of diseases for oncology), direct comparisons with the WHO 2021 classification are extremely challenging. The report demonstrates that astrocytomas (40.5%), intracranial and intraspinal embryonal tumors (20.7%), other gliomas (10.9%), and ependymomas and choroid plexus tumors (10.2%) were the most common tumor families among pediatric patients [19]. Similarly, the VIGICANCER report, which collects childhood cancer statistics across 10 Colombian cities, suffers from a similar limitation in comparison as it bases its grouping on the International Classification of Childhood Cancer guidelines [17].

With respect to the regional literature, a Saudi Arabian 10-year report of the National Neurologic Institute indicates that meningiomas and glioblastomas were the most common among adults [20]. Conversely, low-grade gliomas and medulloblastomas were the most common among pediatrics. A report describing the epidemiology of CNS tumors in Western Saudi Arabia concluded that glioblastoma and meningiomas were the most common tumors across a combined sample of adults and pediatrics [21]. An 11-year reporting on Lebanese patients with CNS tumors demonstrates that meningiomas and glioblastomas were the most common among adults [22]. On the other hand, embryonal tumors, glioblastomas, and meningiomas were the most prevalent among pediatrics. A 5-year report from Ain Shams University in Egypt showed similar results to the Lebanese study in the adults cohort [23]. Interestingly, among the Egyptian pediatric sample, gliomas were the most prevalent while meningioma only comprised 0.9% of cases. Finally, a Moroccan charter of 543 pediatric patients demonstrated that embryonal tumors (i.e., medulloblastoma) and gliomas were the most prevalent tumors [24]. Across that sample, meningiomas only comprised 2.2% of cases. Interestingly, all of the aforementioned reports did not include any statistics on survival outcomes.

Medulloblastomas display some interesting heterogeneity across literature. Its mortality rate ranges anywhere from 15 to 70%. This wide range of prognosis is owed to differences in age at diagnosis, presentation at diagnosis (i.e., presence of metastasis), and tumor histology [25,26]. Local reports from Morocco and Saudi Arabia estimated the 5-year survival of medulloblastoma at 52% in the Moroccan cohort, and 58% for WNT medulloblastoma and 65% for SHH medulloblastoma across the Saudi cohort [27,28]. A Jordanian report of pediatric patients with medulloblastoma estimates 5-year overall survival at 69.4% [29]. Survival was highest for patients with the SHH molecular variant (77.8%) and lowest for group-3 medulloblastoma (41.4%). Among children between the ages of 0 and 14, the CBTRUS reports a 5-year survival rate of 48% [15], which further dwindles to 11% and 2% in the adolescents and young adults, and the adults groups, respectively. The EUROCARE-5 pediatric report shows 5-year survival rates of 65%, 36%, and 72% for medulloblastoma variants, large cell medulloblastoma, and desmoplastic/nodular medulloblastoma, respectively [19]. Finally, the VIGICANCER report shows the 5-year survival of medulloblastoma at 61% [17].

Finally, we demonstrated that IDH-mutant gliomas have superior survival compared to their wild-type variants. Such findings were consistently observed during retrospective as well as clinical studies and rendered IDH mutation status an important factor to consider for treatment planning [30]. Studies have shown that IDH alone is insufficient to risk stratifying certain gliomas, such as astrocytic tumors [31]. Some researchers advocate for the use of a combination of molecular biomarkers (e.g., ATRX, p53, TERT) to sub-classify gliomas as a whole [32,33,34,35]. However, the aforementioned only serves to augment the resource-intensive nature of pursuing such markers, irrespective of their clinical utility.

Collectively, the abovementioned shows three distinct disparities in neuro-oncology. The first, which is the most obvious, is the heterogeneous grouping of tumor entities due to the differences in utilized guidelines. The new WHO guidelines of 2021 simplified the grouping of tumor entities and abolished many categories that are now considered obsolete. The second issue is the scarcity of detailed, large-scale reporting on CNS tumors for a variety of regions, particularly South America, the Middle East and North Africa, and West Asia. In fact, a plethora of region-based reports on CNS cancers are mere sub-groupings of the Global Burden of Disease estimates for an extremely vague category— “Brain and central nervous system cancer” [2,36]. Finally, the most significant contributor to disparity in neuro-oncology could be the systemic barriers and/or differences present within each region. These include economic instability, quality of education, quality of cancer care services, and social/cultural differences [37].

A pertinent finding of our paper is the presence of sex-based differences in survival across the adult but not pediatric cohort with CNS tumors. From an epidemiological perspective, males are subjected to significantly greater excess of CNS cancer burden including incidence and mortality; the latter ranges from 20 to 30% [38]. Moore et al., in their analysis of the National Cancer Database, demonstrated that males have worse survival across all types of CNS tumors as compared to females [8,38]. On the other hand, the biological perspective indicates that sex affects brain tumors at multiple stages or conditions, which include tumor initiation, tumor growth, and tumor response to treatment [38]. Moreover, the literature demonstrates that males have a higher number of cells; thus, they are predisposed to more mitotic events, which leads to an increased likelihood of acquiring somatic mutations [39]. Other studies advocated for the role of sex-specific hormones (e.g., protective effects of estrogen) on the impact on CNS tumor burden [40,41]. Moreover, it appears that the incomplete inactivation of the female X chromosome might be protective for females [42].

We also demonstrated that pediatric patients have favorable survival rates compared to their adult counterparts. This finding is well in-line with the literature and our understanding of the dynamics of cancer in general. Age is a known risk factor for cancer and is associated with dismal mortality for an array of different tumors [43]. Increased age is also associated with the development of multiple comorbidities [44]. Moreover, it is associated with a significant decrease in immune-system effectiveness [45,46,47,48]. Ladomerksy et al. theorize that the mere process of normal human aging progressively suppresses immunosurveillance, which acts as a primary mechanism of the initiation of tumors, specifically glioblastoma, later in life [43]. However, due to the significant differences in the epidemiology of CNS tumors and their associated treatment between adults and pediatrics, it is best to analyze CNS tumors separately for each age strata [49].

Our findings, while mirroring the trends of CNS tumors across the literature, reiterate the fact that the burden of CNS tumors in the MENA region is rising. This is mainly attributed to increased life expectancy, urban expansion, lifestyle changes, and the adoption of more advanced diagnostic and treatment modalities [50]. Challenges to proper policy making across the MENA region, across which the KHCC is a major player, include lack of proper screening, lack of specialized centers, scarcity of resources, and the high epidemiological burden of CNS cancers. On the surface level, CNS cancers exhibit great variability across regions, age groups, and affected biological sexes. Thus, policies targeting these cancers should be sensitive, at least, to the aforementioned contexts and their associated clinical implications.

During the study period (2003–2019), several novel therapies for CNS tumors were introduced, including anti-VEGF agents such as bevacizumab and regorafenib, temozolomide (TMZ) in combination with radiotherapy, and tumor-treating fields (TTF) [51,52,53,54]. While specific therapy data were not available in our study, these advancements may have influenced survival outcomes, particularly among adults. Therapies such as TMZ and anti-angiogenic drugs have demonstrated differing efficacy and safety profiles between adults and pediatric patients, reflecting variations in tumor biology and treatment tolerability [55]. For instance, while TMZ is a well-established therapy for adult glioblastomas, its role in pediatric high-grade gliomas is less clearly defined [56]. This disparity may partially explain the significant survival differences observed between adult and pediatric patients in our cohort. Future studies incorporating detailed therapy information are necessary to evaluate the impact of these therapeutic advancements on survival outcomes.

This paper is not devoid of limitations. Firstly, it has all the inherent biases of a retrospective chart review. Secondly, the WHO 2021 classification (CNS5) was applied retrospectively to cases based on histopathological and radiological reports. No molecular analysis was applied to further verify cases. Thirdly, only a handful of characteristics of extracted and analyzed variables are for clinical impact. Fourthly, follow-up was not available for all cases for all study time points. Fifthly, the precision of some survival estimates might be reduced by the low sample sizes for some CNS groups and entities. Sixthly, survival differences were not studied within the context of clinically relevant variables (e.g., treatment). We acknowledge that this is an institution-based report and may not fully represent the entire region. However, KHCC treats 60–70% of cancer patients in Jordan, and national cancer registries do not provide outcome measurements, making KHCC the most suitable center for assessing CNS tumors in Jordan. Differences in access to healthcare, availability of specialized treatments, and socioeconomic factors may result in variations in CNS tumor epidemiology and outcomes in other settings. Nonetheless, KHCC’s position as one of the largest and most specialized cancer centers in the region strengthens the validity of our findings as representative of the trends in Jordan. Future multi-center studies and regional collaborations are necessary to validate our results and explore potential variations in CNS tumor outcomes across different populations and healthcare systems.

Finally, due to the heterogeneity of classification and reporting across the literature, comparisons might exhibit an innate degree of error.

## 5. Conclusions

In conclusion, our study offers a detailed evaluation of 2094 patients treated between 2003 and 2019. The majority of tumors were gliomas, glioneuronal, and neuronal tumors, followed by meningiomas and embryonal tumors. Survival outcomes showed significant disparities, with pediatric patients having a median survival of 266 months compared to 76 months for adults. Males had worse outcomes than females, with median survivals of 71 and 139 months, respectively. We also found out that adult-type diffuse gliomas and pediatric-type diffuse high-grade gliomas exhibited the poorest survival rates. Additionally, the presence of IDH mutations in adult gliomas was associated with better outcomes compared to wildtype cases. These findings align with global trends but underscore the rising burden of CNS tumors in Jordan. This study spotlights the need for region-specific cancer policies, improved diagnostic tools, and treatment approaches to address sex-based and age-based differences.

## Figures and Tables

**Figure 1 cancers-17-00590-f001:**
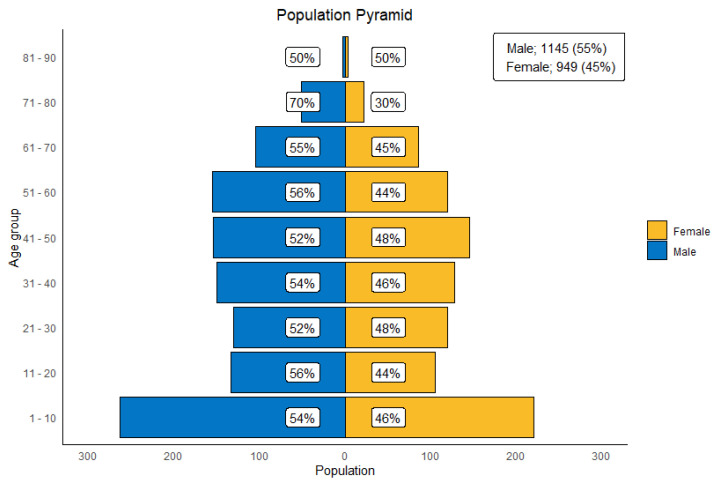
Population pyramid representing the age and gender distribution of patients with primary Central Nervous System (CNS) tumors managed at KHCC between July 2003 and June 2019.

**Figure 2 cancers-17-00590-f002:**
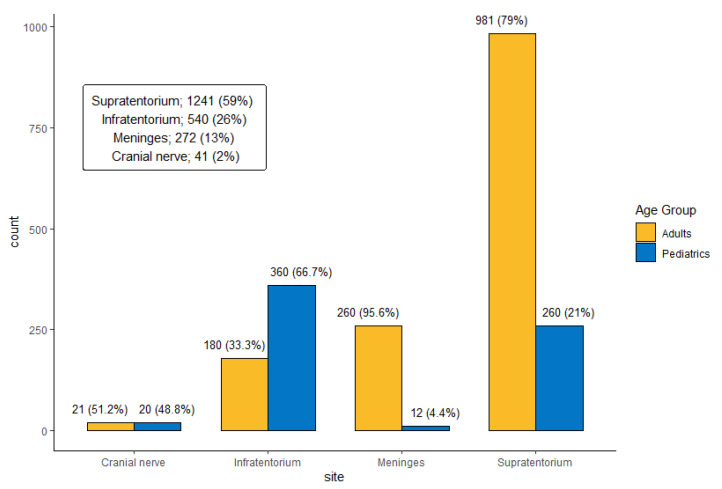
Distribution of primary Central Nervous System (CNS) tumors by anatomical site and age group at KHCC between July 2003 and June 2019.

**Figure 3 cancers-17-00590-f003:**
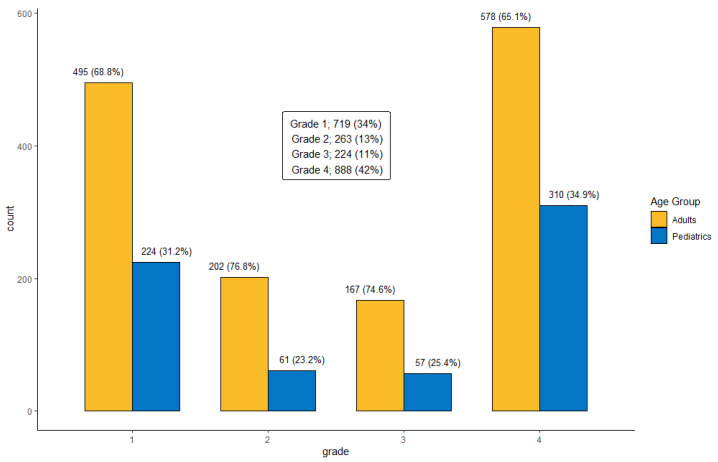
Distribution of primary Central Nervous System (CNS) tumors by WHO grade and age group at KHCC between July 2003 and June 2019.

**Figure 4 cancers-17-00590-f004:**
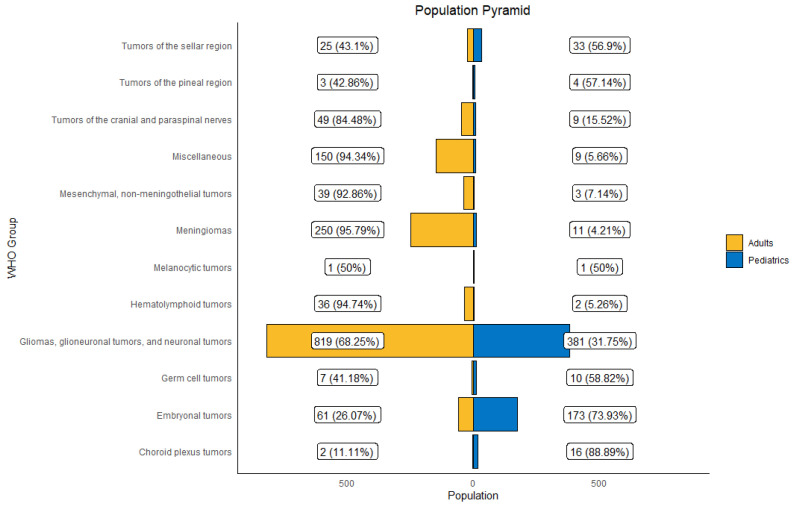
Distribution of primary Central Nervous System (CNS) tumors by WHO tumor group and age group at KHCC between July 2003 and June 2019.

**Figure 5 cancers-17-00590-f005:**
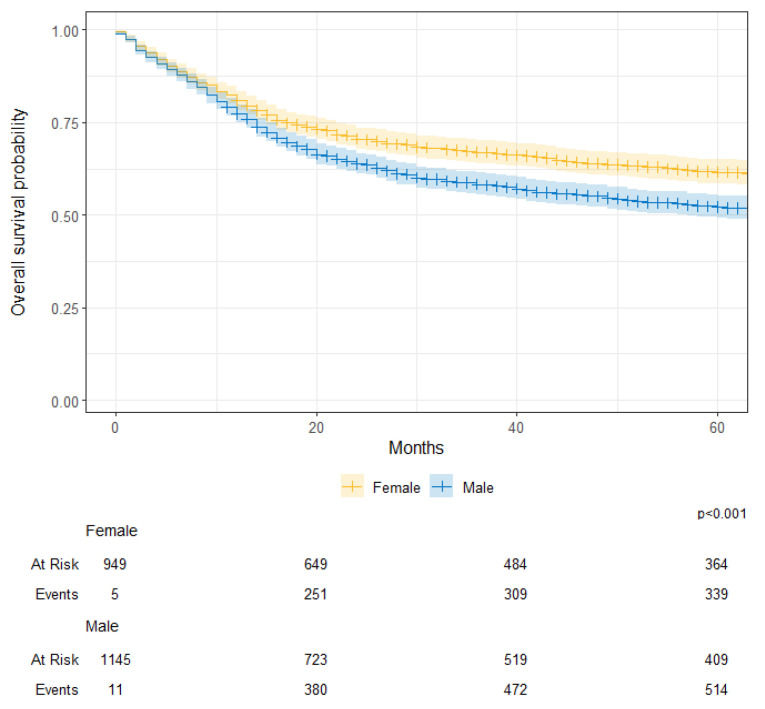
Overall survival probability by gender (male vs. female) for patients with primary Central Nervous System (CNS) tumors at KHCC between July 2003 and June 2019.

**Figure 6 cancers-17-00590-f006:**
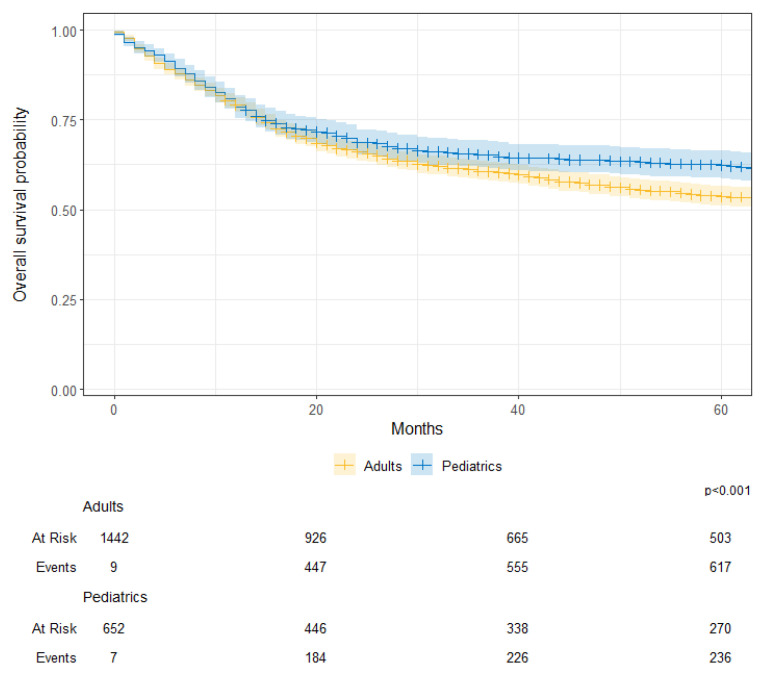
Overall survival probability by age group (adults vs. pediatrics) for patients with primary Central Nervous System (CNS) tumors at KHCC between July 2003 and June 2019.

**Figure 7 cancers-17-00590-f007:**
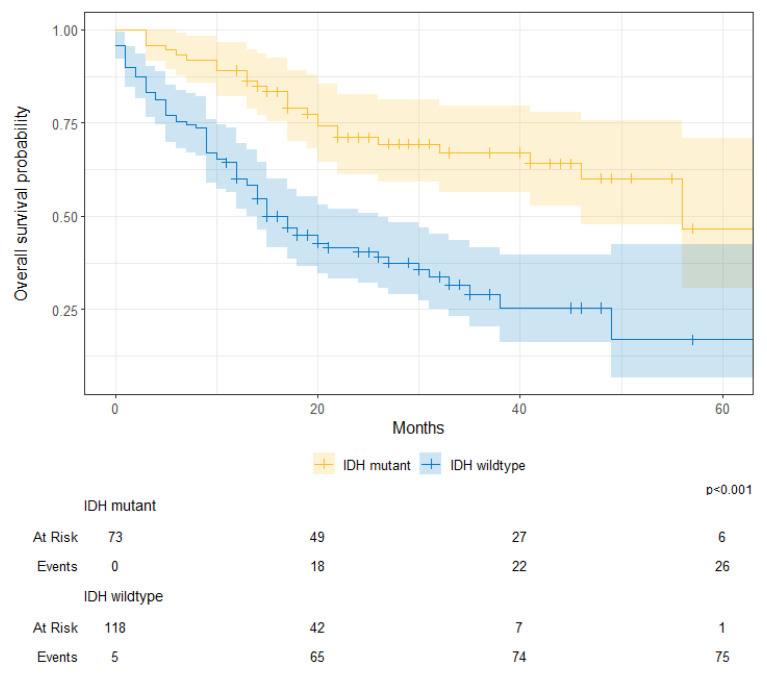
Overall survival probability for patients with IDH-mutant and IDH-wildtype gliomas at KHCC between July 2014 and June 2019.

**Figure 8 cancers-17-00590-f008:**
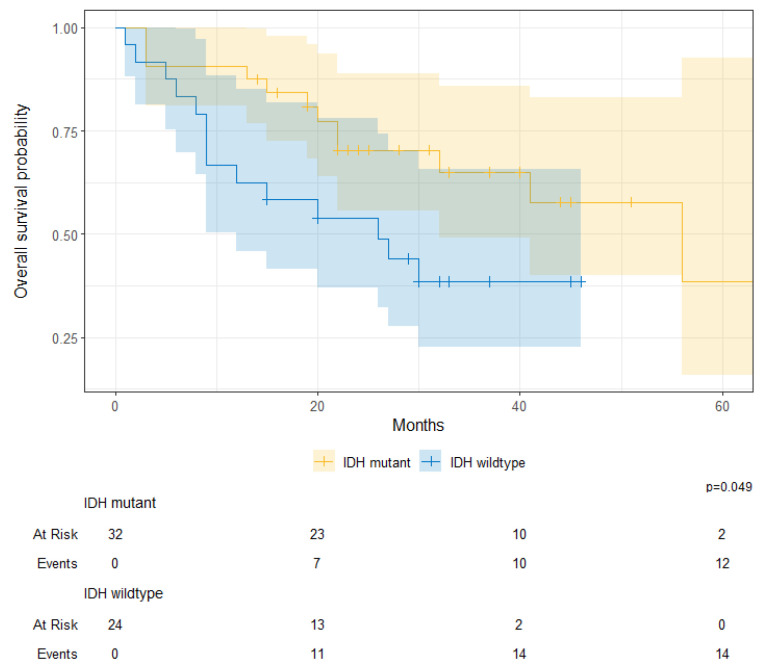
Overall survival probability for patients with IDH-mutant and IDH-wildtype astrocytomas at KHCC between July 2014 and June 2019, irrespective of grade.

**Figure 9 cancers-17-00590-f009:**
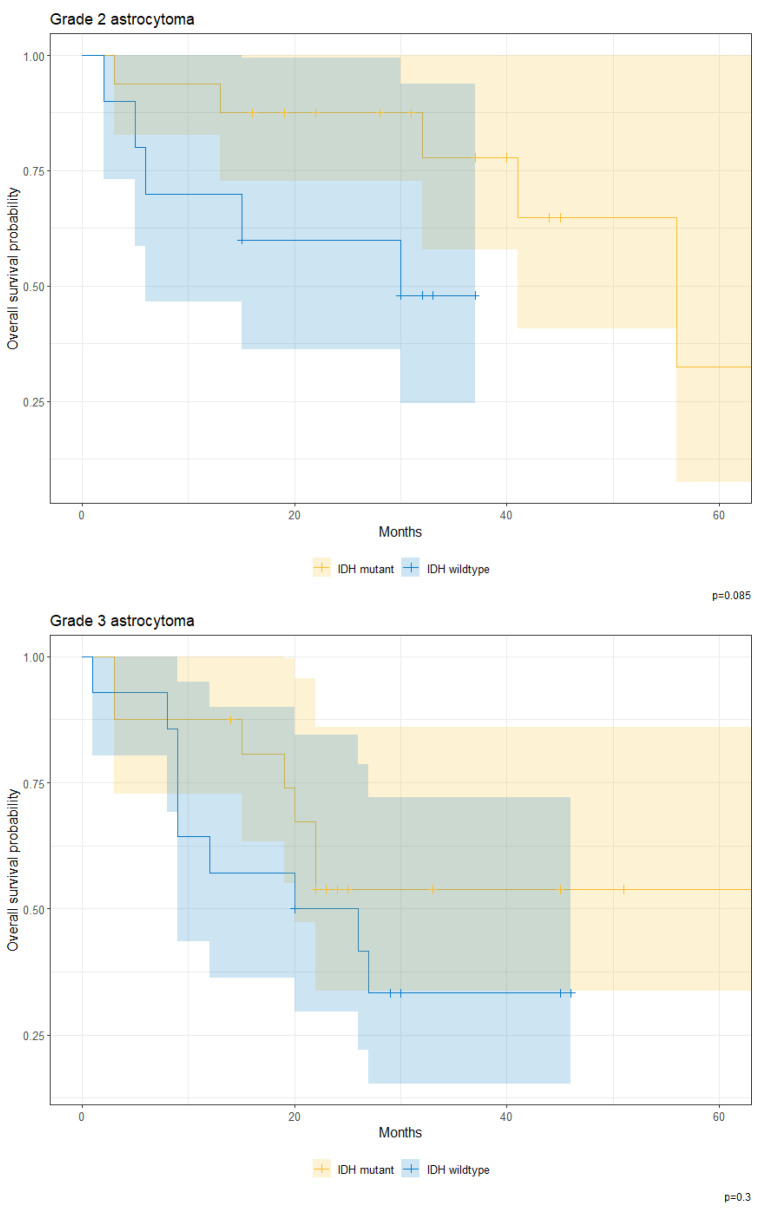
Overall survival probability for patients with IDH-mutant and IDH-wildtype astrocytomas sorted by grade (grade 2 and 3) at KHCC between July 2014 and June 2019.

**Figure 10 cancers-17-00590-f010:**
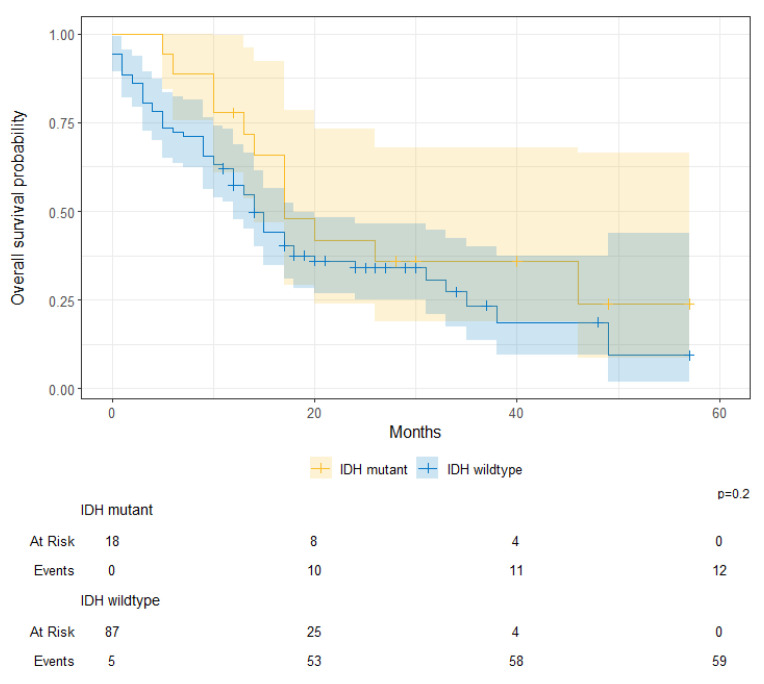
Overall survival probability for patients with IDH-mutant and IDH-wildtype glioblastoma at KHCC between July 2014 and June 2019.

**Figure 11 cancers-17-00590-f011:**
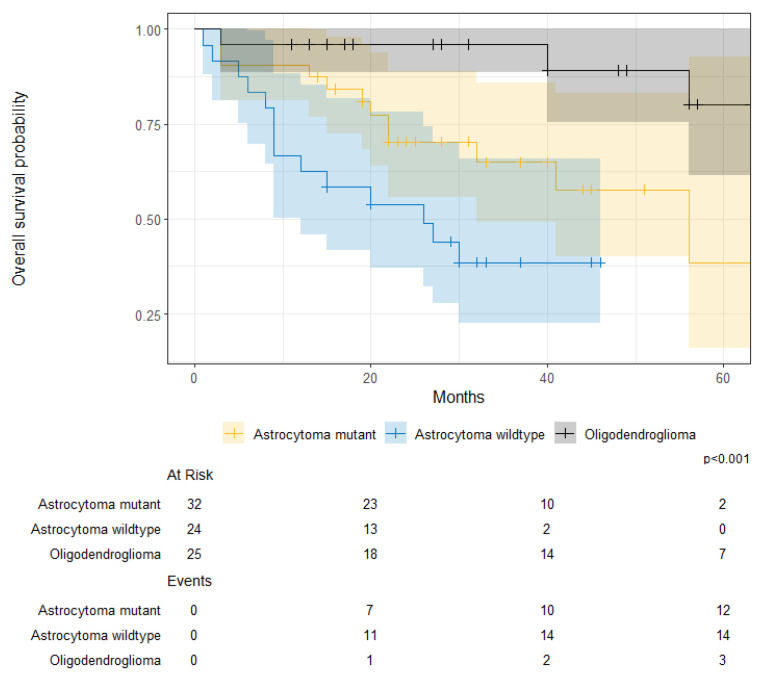
Overall survival probability for patients with astrocytoma mutant, astrocytoma wildtype, and oligodendroglioma between July 2014 and June 2019.

**Figure 12 cancers-17-00590-f012:**
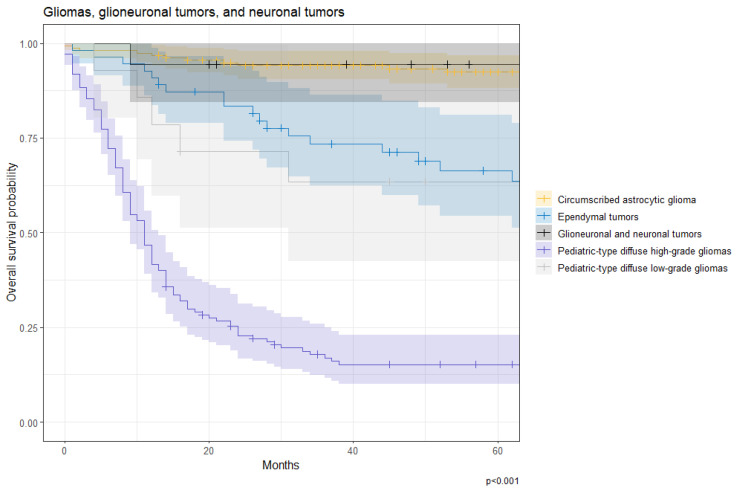
Overall survival probability for patients with gliomas, glioneuronal tumors, and neuronal tumors by subtype at KHCC between July 2003 and June 2019.

**Table 1 cancers-17-00590-t001:** Distribution of primary Central Nervous System (CNS) tumors by WHO classification, gender, and male-to-female ratio in patients managed at KHCC between July 2003 and June 2019.

		Female		Male		
WHO Classification	Total	N	%	N	%	Ratio
Adult-type diffuse gliomas	709	270	38.1	439	61.9	1.62
Pediatric-type diffuse low-grade gliomas	14	4	28.6	10	71.4	2.5
Pediatric-type diffuse high-grade gliomas	162	74	45.7	88	54.3	1.19
Circumscribed astrocytic glioma	183	87	47.5	96	52.5	1.10
Glioneuronal and neuronal tumors	43	19	44.2	24	55.8	1.26
Ependymal tumors	89	42	47.2	47	52.8	1.12
Choroid plexus papilloma	4	3	75	1	25	0.3
Atypical choroid plexus papilloma	8	3	37.5	5	62.5	1.7
Choroid plexus carcinoma	6	0	0	6	100	∞
Medulloblastoma	199	75	37.7	124	62.3	1.7
Atypical teratoid/rhabdoid tumor	12	7	58.3	5	41.7	0.71
Embryonal tumor with multilayered rosettes	6	4	66.7	2	33.3	0.5
CNS embryonal tumor	17	10	58.8	7	41.2	0.7
Pineoblastoma	6	1	16.7	5	83.3	5
Papillary tumor of the pineal region	1	1	100	0	0	0.0
Germinoma	15	2	13.3	13	86.7	6.5
Teratoma	1	0	0	1	100	∞
Choriocarcinoma	1	1	100	0	0	0.0
Meningioma	213	146	68.5	67	31.5	0.46
Atypical meningioma	30	18	60	12	40	0.67
Anaplastic meningioma	18	10	55.6	8	44.4	0.8
Neurofibroma	51	25	49	26	51	1.04
Malignant peripheral nerve sheath tumor	7	4	57.1	3	42.9	0.75
Solitary fibrous tumor/hemangiopericytoma	11	5	45.5	6	54.5	1.2
Meningeal melanoma	2	2	100	0	0	0.0
Pituitary neuroendocrine tumors (PitNETs)	157	75	47.8	82	52.2	1.1
Pituicytoma	2	0	0	2	100	∞
Craniopharyngioma	56	31	55.4	25	44.6	0.81
Hemangioblastoma	16	8	50	8	50	1.0
Dermoid cyst	2	1	50	1	50	1.0
Lipoma	2	0	0	2	100	∞
Hemangioma	13	4	30.8	9	69.2	2.25
Diffuse large B-cell lymphoma of the CNS	38	17	44.7	21	55.3	1.23

**Table 2 cancers-17-00590-t002:** Anatomic distribution of primary Central Nervous System (CNS) tumors by WHO classification and location at KHCC between July 2003 and June 2019.

	Infratentorium	Supratentorium	Meninges	Cranial Nerve
WHO Classification	N	N	N	N
Adult-type diffuse gliomas	15	694	0	0
Anaplastic meningioma	0	0	18	0
Atypical choroid plexus papilloma	0	8	0	0
Atypical meningioma	0	0	30	0
Atypical teratoid/rhabdoid tumor	7	5	0	0
Choriocarcinoma	0	1	0	0
Choroid plexus carcinoma	0	6	0	0
Choroid plexus papilloma	0	4	0	0
Circumscribed astrocytic glioma	82	83	0	18
CNS embryonal tumor	5	12	0	0
Craniopharyngioma	0	56	0	0
Dermoid cyst	1	1	0	0
Diffuse large B-cell lymphoma of the CNS	8	30	0	0
Embryonal tumor with multilayered rosettes	1	5	0	0
Ependymal tumors	67	22	0	0
Germinoma	0	15	0	0
Glioneuronal and neuronal tumors	8	35	0	0
Hemangioblastoma	10	6	0	0
Hemangioma	7	6	0	0
Lipoma	0	0	2	0
Malignant peripheral nerve sheath tumor	4	2	0	1
Medulloblastoma	199	0	0	0
Meningeal melanoma	2	0	0	0
Meningioma	1	1	211	0
Neurofibroma	19	10	0	22
Papillary tumor of the pineal region	0	1	0	0
Pediatric-type diffuse high-grade gliomas	97	65	0	0
Pediatric-type diffuse low-grade gliomas	6	8	0	0
Pineoblastoma	0	6	0	0
Pituicytoma	0	2	0	0
Pituitary neuroendocrine tumors (PitNETs)	0	157	0	0
Solitary fibrous tumor/hemangiopericytoma	0	0	11	0
Teratoma	1	0	0	0

**Table 3 cancers-17-00590-t003:** Survival rates at 1, 2, and 5 years for primary Central Nervous System (CNS) tumors for the adult cohort including the WHO groups, and the 2 most common tumor families (gliomas, glioneuronal and neuronal tumors, and meningiomas).

Characteristic	1-Year	2-Year	5-Year
WHO Group			
Choroid plexus tumors	100% (100%, 100%)	100% (100%, 100%)	100% (100%, 100%)
Embryonal tumors	98% (95%, 100%)	92% (85%, 99%)	70% (59%, 83%)
Germ cell tumors	100% (100%, 100%)	100% (100%, 100%)	100% (100%, 100%)
Gliomas, glioneuronal tumors, and neuronal tumors	67% (64%, 70%)	47% (43%, 50%)	33% (29%, 36%)
Hematolymphoid tumors	64% (50%, 82%)	46% (32%, 66%)	24% (12%, 48%)
Melanocytic tumors	— (—, —)	— (—, —)	— (—, —)
Meningiomas	98% (96%, 100%)	95% (92%, 97%)	84% (78%, 89%)
Mesenchymal, non-meningothelial tumors	90% (81%, 100%)	87% (77%, 98%)	83% (71%, 97%)
Miscellaneous	98% (96%, 100%)	97% (95%, 100%)	92% (87%, 97%)
Tumors of the cranial and paraspinal nerves	100% (100%, 100%)	98% (94%, 100%)	90% (82%, 100%)
Tumors of the pineal region	67% (30%, 100%)	33% (6.7%, 100%)	33% (6.7%, 100%)
Tumors of the sellar region	96% (89%, 100%)	96% (89%, 100%)	87% (74%, 100%)
Gliomas, glioneuronal tumors, and neuronal tumors			
Adult-type diffuse gliomas	64% (61%, 68%)	42% (39%, 46%)	28% (24%, 31%)
Circumscribed astrocytic glioma	81% (67%, 97%)	81% (67%, 97%)	81% (67%, 97%)
Ependymal tumors	97% (92%, 100%)	87% (76%, 100%)	73% (58%, 91%)
Glioneuronal and neuronal tumors	88% (76%, 100%)	88% (76%, 100%)	83% (69%, 100%)
Pediatric-type diffuse high-grade gliomas	72% (56%, 92%)	45% (29%, 71%)	38% (21%, 67%)
Meningiomas			
Anaplastic meningioma	94% (84%, 100%)	94% (84%, 100%)	66% (45%, 96%)
Atypical meningioma	96% (89%, 100%)	92% (82%, 100%)	83% (69%, 100%)
Meningioma	98% (96%, 100%)	95% (92%, 98%)	85% (80%, 91%)

**Table 4 cancers-17-00590-t004:** Survival rates at 1, 2, and 5-years for primary Central Nervous System (CNS) tumors for the pediatric cohort with including the WHO groups, and the 2 most common tumor families (gliomas, glioneuronal and neuronal tumors, and embryonal tumors).

Characteristic	1-Year	2-Year	5-Year
WHO Group			
Choroid plexus tumors	100% (100%, 100%)	87% (72%, 100%)	72% (52%, 100%)
Embryonal tumors	77% (71%, 84%)	65% (58%, 72%)	58% (51%, 67%)
Germ cell tumors	100% (100%, 100%)	90% (73%, 100%)	80% (59%, 100%)
Gliomas, glioneuronal tumors, and neuronal tumors	75% (71%, 80%)	66% (62%, 71%)	60% (55%, 65%)
Hematolymphoid tumors	50% (13%, 100%)	0% (13%, 100%)	— (—, —)
Melanocytic tumors	100% (100%, 100%)	100% (100%, 100%)	— (—, —)
Meningiomas	91% (75%, 100%)	91% (75%, 100%)	91% (75%, 100%)
Mesenchymal, non-meningothelial tumors	67% (30%, 100%)	67% (30%, 100%)	67% (30%, 100%)
Tumors of the cranial and paraspinal nerves	100% (100%, 100%)	100% (100%, 100%)	100% (100%, 100%)
Tumors of the pineal region	100% (100%, 100%)	50% (19%, 100%)	50% (19%, 100%)
Tumors of the sellar region	91% (82%, 100%)	85% (73%, 98%)	75% (60%, 94%)
Gliomas, glioneuronal tumors, and neuronal tumors			
Circumscribed astrocytic glioma	97% (94%, 100%)	94% (91%, 98%)	92% (88%, 97%)
Ependymal tumors	91% (84%, 99%)	83% (74%, 94%)	66% (54%, 81%)
Glioneuronal and neuronal tumors	94% (84%, 100%)	94% (84%, 100%)	94% (84%, 100%)
Pediatric-type diffuse high-grade gliomas	42% (34%, 51%)	23% (17%, 31%)	15% (10%, 23%)
Pediatric-type diffuse low-grade gliomas	79% (60%, 100%)	71% (51%, 99%)	63% (42%, 95%)
Embryonal tumors			
Atypical teratoid/rhabdoid tumor	25% (9.4%, 67%)	17% (4.7%, 59%)	— (—, —)
CNS embryonal tumor	80% (59%, 100%)	20% (5.8%, 69%)	20% (5.8%, 69%)
Embryonal tumor with multilayered rosettes	33% (11%, 100%)	17% (2.8%, 100%)	— (—, —)
Medulloblastoma	83% (78%, 90%)	74% (67%, 81%)	67% (59%, 75%)

## Data Availability

The data that supports the findings of this study are available from the corresponding author.

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
