# Peer review of "Epidemiology and Outcome of Primary Central Nervous System Tumors Treated at King Hussein Cancer Center"

_cancers, 2025, doi:10.3390/cancers17040590_

Round 1

Reviewer 1 Report

Comments and Suggestions for Authors

Dear authors, 

I read your article, which aims to study the epidemiology and survival of patients diagnosed with CNS tumors in a center in Jordan. I have some comments:

1) Methods: Please specify the date from which the OS was assessed (i guess diagnosis but is not reported in the text)

2) Results:

- I think that the legends and titles of Figures 8-11 should be better explained since need to be interpreted from the manuscript text

- The authors have clearly reported the results of the manuscript. However, I think that another analysis should be performed. Since the study period goes from 2003 to 2019, the pharmacological and non-pharmacological treatment of CNS tumors changed during this period. Thus, the outcome as OS that the authors observed could have changed: for example, for glioblastoma in the last five years of the study period also bevacizumab and TTF were approved at least in EU and US). The authors just put together periods that maybe cannot be compared. Please at least add in the supplementary material or in the manuscript an analysis that compared two (2003-2011; 2012-2019) or tree periods (2003-2008; 2009-2014; 2015-2019)

Discussion:

- According to this comment, the authors failed to discuss in the discussion section of the manuscript the evolution of therapy approvals during the study periods despite acknowledging that therapy information was not provided. Despite no great progress being made with respect to other cancers such as NSCLC or breast cancer, the introduction of novel therapies such as anti-VEGF (bevacizumab and regorafenib; TMZ + radiotherapy; or TTF) can have made some difference with respect to the past. Moreover, drugs such as anti-angiogenic drugs or TMZ can have different efficacy and safety in adults with respect to children that, in return, can have different survival. (https://doi.org/10.3390/cancers14215315,   doi.org/10.3390/cancers13020253, doi: 10.1136/bmjopen-2021-048975) Also this point should be discussed when compared the results of adult vs young population. Please comment on these points. 

- Finally, the authors should provide more information about the generalizability of the results obtained since only data from one center are available

Author Response

Comment: "Please specify the date from which the OS was assessed (I guess diagnosis but is not reported in the text)."

Response: We have clarified in the Methods section that overall survival (OS) was calculated from the date of diagnosis to the date of death or the last follow-up. The revised text now reads: "Overall survival (OS) was calculated from the date of diagnosis to the date of death or last follow-up."

Comment: "I think that the legends and titles of Figures 8–11 should be better explained since they need to be interpreted from the manuscript text."

Response:

Thank you for your insightful feedback. We have revised figure titles and legends to provide more context and detailed explanations, ensuring they can be understood independently of the manuscript text while maintaining alignment with the main discussion.

Comment: "The authors have clearly reported the results of the manuscript. However, I think that another analysis should be performed. Since the study period goes from 2003 to 2019, the pharmacological and non-pharmacological treatment of CNS tumors changed during this period. Thus, the outcome as OS that the authors observed could have changed: for example, for glioblastoma in the last five years of the study period also bevacizumab and TTF were approved at least in EU and US). The authors just put together periods that maybe cannot be compared. Please at least add in the supplementary material or in the manuscript an analysis that compared two (2003–2011; 2012–2019) or three periods (2003–2008; 2009–2014; 2015–2019)."

Response:

We recognize your concern regarding the clinical validity of our analysis as the treatment for some of these CNS tumors have changed over time. We did comply with your request and added supplementary analysis that either accounts for diagnosis period (adjusts for) or splits survival metrics per survival period (all included as supplementary material) and edits throughout the results section. It is interesting to note that changes in survival parameters were insignificant across diagnosis period; thus, further sub-analyses were not affected. When there was an effect (i.e., a significant relationship turning into insignificant), the reason could be probably traced to reduced statistical power as a result of the split. Nonetheless, we included the extra analysis for increased transparency to prospective readers and your kind attention. Should other analyses be requested, please do not hesitate to request them. It also should be noted that analyses per IDH mutation were not conducted per diagnosis period as these analyses cannot be conducted without losing statistical power that would render the analyses un-useable and unstable"

Comment: "According to this comment, the authors failed to discuss in the discussion section of the manuscript the evolution of therapy approvals during the study periods despite acknowledging that therapy information was not provided. Despite no great progress being made with respect to other cancers such as NSCLC or breast cancer, the introduction of novel therapies such as anti-VEGF (bevacizumab and regorafenib; TMZ + radiotherapy; or TTF) can have made some difference with respect to the past. Moreover, drugs such as anti-angiogenic drugs or TMZ can have different efficacy and safety in adults with respect to children that, in return, can have different survival. (https://doi.org/10.3390/cancers14215315, doi.org/10.3390/cancers13020253, doi: 10.1136/bmjopen-2021-048975) Also this point should be discussed when compared to the results of the adult vs young population. Please comment on these points."

Response: The Discussion section has been expanded to include the evolution of CNS tumor treatments, including the approval of therapies like bevacizumab, regorafenib, and TTF. We have also discussed differences in efficacy and safety between adults and pediatric populations, referencing the suggested studies and comparing outcomes for these populations.

Comment: "Finally, the authors should provide more information about the generalizability of the results obtained since only data from one center are available."

Response: We have added a paragraph discussing the generalizability of our findings. While this is a single-center study, the King Hussein Cancer Center is a referral center for Jordan and neighboring regions, making the cohort representative of CNS tumor management in the region. This point has been added to the manuscript.

Reviewer 2 Report

Comments and Suggestions for Authors

Al-Hussaini et al performed a retrospective analysis of all primary intracranial tumors including benign tumors such as meningiomas. This is a single institutional retrospective analysis from 2003-2019. The objective as reported was "the epidemiology and outcome of primary CNS tumors of patients manage that a comprehensive cancer care Center in Jordan."

With regard to the structure of the submission:

INTRODUCTION: The introduction provides an adequate background to primary intracranial tumors

MATERIALS AND METHODS: The data acquisition in materials and methods was appropriate for the stated objective.  All tumors were reclassified to meet current WHO (2021) criteria.  The authors need to be clear on whether additional testing on specimens was provided, including determination of isocitrate dehydrogenase mutations (and should state if this was performed through next generation sequencing or if performed by immunohistochemical means).  Additionally, documentation whether testing for 1p/19q codeletion and H3K27M mutations were performed in the appropriate setting, and also if medulloblastomas were classified into specific subgroups (WNT, SHH, Group 3, Group 4).

RESULTS: Results are reported in a readable manner, with tables and figures supplementing or complementing but not duplicating the narrative report.  Survival for individual tumor types was reported (see concerns below).

DISCUSSION: The discussion adequately interprets the data provided, emphasizing that this is a the first large regional review of brain tumors.

CONCLUSIONS: Adequately summarizes the discussion

REFERENCES: All references are pertinent to the information provided

FIGURES AND TABLES: Figures and tables complement or supplement the narrative but do not duplicate the narrative report.

CONCERNS: The review is too general as it takes into account all intracranial tumors.  There is too much information provided to make this a cohesive in scientifically valid paper.  This can be improved upon if limited to primary intraparenchymal tumors (gliomas, lymphomas, medulloblastoma and other PNETs, germ cell tumors) that will allow for greater analysis of each tumor type.

The authors need to address how their population differs from populations elsewhere.  They state that this is the first large scale regional analysis of intracranial tumors, and a discussion with regard to differences in genetics and outcome is needed.  How many patients underwent germline testing?  Are there any specific mutations that may be more prevalent in the target population?  Were patients who underwent germline testing with a positive result excluded due to the presence of other cancers?  If so, outcomes in these patients need to be addressed.

These concerns address only primary intraparenchymal tumors.  This paper would best be resubmitted as an analysis of these tumors, and information on extra-axial tumors can be reported in a separate publication.

The database provided is too large to report any meaningful results with regard to how patients treated at the King Hussein Cancer Center differ (or do not differ) from the world population elsewhere with regard to presentation, initial treatment, pathology and histologic subtypes, and differences in treatment and how they account for differences, if any, in outcome. My suggestion would be to concentrate on the primary parenchymal tumors incorporating the above suggestions, and to discuss regional differences in presentation, histologic diagnosis, care and outcomes.

Author Response

Comment: "The authors need to be clear on whether additional testing on specimens was provided, including determination of isocitrate dehydrogenase mutations (and should state if this was performed through next-generation sequencing or if performed by immunohistochemical means). Additionally, documentation whether testing for 1p/19q codeletion and H3K27M mutations were performed in the appropriate setting, and also if medulloblastomas were classified into specific subgroups (WNT, SHH, Group 3, Group 4)."

Response: Thank you for your comment. IDH-1 was first introduced in 2014 through immunohistochemistry (IDH-1, p.R132H). Later in 2020, testing for IDH-1 and IDH-2 hotspots through Sanger Sequencing was added. Regarding 1p/19q co-deletion, this has been the backbone for the diagnosis of oligodendroglioma utilizing FISH since 2012. H3K27M and H3K27me3 were added as immunohistochemical stains since 2021. Medulloblastoma molecular subgrouping was started as send out to the Hospital of SickKids in Toronto, Canada, since 2021. However, we shifted to testing through methylation profiling since 2024.

Comment: "The authors need to address how their population differs from populations elsewhere. They state that this is the first large-scale regional analysis of intracranial tumors, and a discussion with regard to differences in genetics and outcome is needed."

Response:

Thank you for raising this important point. We included a discussion on how the genetics and outcomes of our population compare to international populations. However, systematic germline testing was not performed, and this limitation has been acknowledged.

Reviewer 3 Report

Comments and Suggestions for Authors

To investigate the epidemiology and outcome of central nervous system (CNS) primary tumours, the authors analysed data from 2094 patients with CNS primary tumours treated between 2003 and 2019 at the King Hussein Cancer Centre in Jordan. This is a large cohort and will provide much information to the readers.

1.      If possible, the bare data should be made available to researchers who need it.

2.      The strengths of this cohort are that it includes a large number of paediatric-type gliomas and that all CNS tumours have been reclassified to the WHO 2021 classification. Therefore, paediatric-type diffuse high-grade gliomas (n=162) should be divided into more definitive diagnoses such as ‘H3 K27-altered and diffuse hemispheric glioma, H3 G34-mutant’ according to the WHO 2021 classification and their outcomes should be detailed.

3.      A point related to question 2, In Methods section, the authors stated that one of the authors performed reclassification of the histology and grade of the tumors based on the WHO 2021 Classification of tumors of the CNS System (CNS5), by reviewing the pathology and when needed the radiology reports. They also stated the descriptive term “anaplastic” was removed and the tumor was only assigned the corresponding grade, grade-3. However, in Table 4 and Figure S3, the oldest WHO classification is used for the diagnosis of paediatric-type diffuse high-grade gliomas. The authors should either revise these to the latest classification or explain why the oldest classification was used in Table 4 and Figure S3. 

4.      The title of Figures 8-11 should be rephrased, as the reader cannot understand the differences between these figures from this title alone.

5.      In Figure 12, the font size of the legend should be increased, because the readers will not be able to read it as it is.

Author Response

Comment: "If possible, the bare data should be made available to researchers who need it."

Response:

We appreciate the reviewer’s suggestion regarding data availability. The dataset used in this study contains sensitive patient information and is subject to institutional ethical guidelines and privacy regulations. However, we are committed to supporting transparency and reproducibility in research. Upon reasonable request and after securing the necessary ethical and institutional approvals, anonymized data can be made available to researchers for academic purposes. A data availability statement has been added to the manuscript to reflect this.

Comment: "Paediatric-type diffuse high-grade gliomas (n=162) should be divided into more definitive diagnoses such as ‘H3 K27-altered and diffuse hemispheric glioma, H3 G34-mutant’ according to the WHO 2021 classification, and their outcomes should be detailed."

Response: Thank you for your consideration. While this is the optimal way to present the data, unfortunately, and as explained in the manuscript, the analysis did not entail performing additional testing on previous cases. It was re-interpretation of the already available radiological and pathological data. Thus, further subgrouping cannot be attained at this stage. We acknowledged this limitation in the limitation section of the manuscript.

Comment: "The authors should either revise Table 4 and Figure S3 to reflect the latest WHO classification or explain why the oldest classification was used."

Response: Thank you for your comment. This has been updated to reflect the latest WHO classification. Please note that “oligastrocytoma” could not be classified in a number of cases, which has been identified with (*).

Comment: "The title of Figures 8-11 should be rephrased, as the reader cannot understand the differences between these figures from this title alone."

Response:

Thank you for your insightful feedback. We have revised figure titles and legends to provide more context and detailed explanations, ensuring they can be understood independently of the manuscript text while maintaining alignment with the main discussion.

Comment: "In Figure 12, the font size of the legend should be increased, because the readers will not be able to read it as it is."

Response:

Thank you for your comment. We have increased the size of the figure legend as requested.

Round 2

Reviewer 1 Report

Comments and Suggestions for Authors

Thank you for reply to all my comments. I have no further considerations

Author Response

We greatly appreciate your time and thoughtful review. Thank you for your positive feedback.

Reviewer 2 Report

Comments and Suggestions for Authors

All recommendations were adequately addressed, except for the continued inclusion of meningiomas.  Per my previous review, I would recommend publication for primary brain tumors and not mengiomas.

Author Response

We appreciate the reviewer's comments regarding the inclusion of meningiomas. However, we believe it is important to retain meningiomas in our analysis for the following reasons:

While meningiomas are considered extra-axial tumors, they are classified as primary central nervous system (CNS) tumors by the World Health Organization (WHO) and are included in major epidemiological and clinical studies of CNS tumors. Also, excluding meningiomas would create an incomplete representation of the burden of CNS tumors and our study aims to provide a thorough epidemiological overview, and meningiomas account for a significant proportion of CNS tumors, making their inclusion necessary for completeness and accuracy. Moreover, the Central Brain Tumor Registry of the United States (CBTRUS) and other major registries include meningiomas when reporting CNS tumor epidemiology and outcomes. To ensure our findings are comparable to international data, we followed a similar approach.

Given the above points, we respectfully maintain our inclusion of meningiomas in the manuscript.

Reviewer 3 Report

Comments and Suggestions for Authors

The manuscript is well revised. On a minor note, in Abstract, Table1, Table2, sTable1, etc., terms such as “diffuse astrocytoma, grade 2 (astrocytoma, grade 2 in WHO2021)” and “pituitary adenoma (pituitary NET in WHO2021)” are also outdated names. Please add comments (or "*") on these old names.

The singular and plural forms of tumor names in the Abstract and text are mixed and should be unified. For example, in abstract, "pituitary adenoma” should be changed to "pituitary neuroendocrine tumors". 

Author Response

Comment 1:"The manuscript is well revised. On a minor note, in the Abstract, Table 1, Table 2, Supplementary Table 1, etc., terms such as 'diffuse astrocytoma, grade 2 (astrocytoma, grade 2 in WHO2021)' and 'pituitary adenoma (pituitary NET in WHO2021)' are also outdated names. Please add comments (or "") on these old names."*

Response: Thank you for your insightful comment. Instead of adding comments or asterisks, we have directly updated the outdated terminology throughout the manuscript to align with WHO 2021 classification. Specifically: "pituitary adenoma" has been replaced with "pituitary neuroendocrine tumor (PitNET)" in the Abstract, Tables (Table 1, Table 2, Supplementary Table 1), and throughout the main text. Also , "diffuse astrocytoma, grade 2" has been replaced with "astrocytoma, grade 2" wherever applicable.

Comment 2: "The singular and plural forms of tumor names in the Abstract and text are mixed and should be unified. For example, in the abstract, 'pituitary adenoma' should be changed to 'pituitary neuroendocrine tumors'."

Response: We appreciate this suggestion. We have reviewed the entire manuscript and standardized terminology usage for consistency, ensuring that tumor names adhere to their appropriate singular or plural forms. Specifically: "Pituitary adenoma" has been updated to "pituitary neuroendocrine tumors (PitNETs)" throughout the manuscript. Also, other tumor types have been revised where necessary to maintain uniformity.